# The Mechanical Behavior and Enhancement Mechanism of Short Carbon Fiber Reinforced AFS Interface

**DOI:** 10.3390/ma15249012

**Published:** 2022-12-16

**Authors:** Chang Yan, Jiaxu Cai, Kun Xiang, Jinfeng Zhao, Wanqing Lei, Changqing Fang

**Affiliations:** Faculty of Printing, Packaging Engineering and Digital Media Technology, Xi’an University of Technology, Xi’an 710048, China

**Keywords:** aluminum foam sandwich, interfacial reinforcement, mechanical properties, enhancement mechanism

## Abstract

The aluminum foam sandwich (AFS), which perfectly combines the excellent merits of an aluminum foam core and face sheet materials, has extensive and reliable applications in many fields, such as aerospace, military equipment, transportation, and so on. Adhesive bonding is one of the most widely used methods to produce AFS due to its general applicability, simple process, and low cost, however, the bonding interface is known as the weak link and may cause a serious accident. To overcome the shortcomings of a bonded AFS interface, short carbon fiber as a reinforcement phase was introduced to epoxy resin to reinforce the interface adhesion strength of AFS. Single lap shear tests and three-point bending tests were conducted to study the mechanical behavior of the reinforced interface and AFS, respectively. The failure mechanism was studied through a macro- and microanalysis. The result showed that after the reinforcement of carbon fiber, the tangential shear strength of the interface increased by 73.65%. The effective displacement of AFS prepared by the reinforced epoxy resin is 125.95% more than the AFS prepared by the unreinforced epoxy resin. The flexure behavior of the reinforced AFS can be compared with AFS made through a metallurgical method. Three categories of reinforcement mechanisms were discovered: (a) the pull off and pull mechanism: when the modified carbon fiber performed as the bridge, the bonding strength improved because of the pull off and pull out of fibers; (b) adhesion effect: the carbon fiber gathered in the hole edge resulted in epoxy resins being gathered in there too, which increased the effective bonding area of the interface; (c) mechanical self-locking effect: the carbon fiber enhanced the adhesive filling performance of aluminum foam holes, which improved the mechanical self-locking effect of the bonding interface.

## 1. Introduction

Aluminum foam has been effectively developed and extensively researched for many years due to its lightweight, high specific energy absorption capacity, easiness of foaming, and so on [1,2,3]. In order to supplement its low strength and high stiffness properties, aluminum foam sandwich structures (AFSs) with high strength and ductile face sheets were studied far and wide [4]. The core layer of AFS stands up to share deformation and brings down flexural deflection. Further, the face sheet mostly bears the axial load during loading. The excellent merits of aluminum foam core and face sheet were compactly combined in AFS, which led to an extensive and reliable application in many fields, such as construction, marine, aerospace, defense, transportation, and so on [5,6,7].

The interface of AFS is taking a vital role as a ligament to connect different types of materials and their properties. If the interfaces are too weak to combine different materials, the AFS cannot take advantage of the sandwich structure. Accordingly, investigations on the interfacial mechanical behavior of AFS were combined with investigations on the mechanical behavior of integral AFS and almost focused on three aspects: (a) quasistatic, (b) dynamic, and (c) other mechanical behavior. So far, quasistatic mechanical behavior has been maturely investigated [8,9]. Zu et al. [10,11] have fabricated AFSs by a powder metallurgy method and an adhesive bonding method and researched their deformations and failure modes under a three-point bending load. The investigation results demonstrated that there were three types of failure modes in the metallurgy bonding sandwich structure: indentation, core shear, and plastic hinge, as well as three types of failure modes in the adhesive bonding sandwich structure: crush, shear damage of the foam core, and delamination of the bonding interface. This was because the metallurgy method produced a firmer bonding interface than the adhesive method. There are five prominent failure modes at a three-point bending load: crush, shear damage of the foam core, fracture of the bottom panel, indentation, and delamination. Different strength analysis models are established for different failure modes [12,13,14,15]. The dynamic mechanical behavior of AFS is also an inescapable element needing to be paid more attention to, as the AFS is generally used to resist impact load in some situations, such as aerospace, shipbuilding, and high-speed rail. The experimental methods for studying the dynamic mechanical behavior of AFSs mainly include the drop weight impact test [15], Split Hopkinson Pressure Bar (SHPB) test [16], gas gun impact test [17], and air explosion test [18]. As a transition, Yu et al. [15] have summarized that quasistatic modes could give an efficient prediction of initial dynamic failure modes of AFS at a lower impact velocity. However, the quasistatic model cannot play to a good effect when it refers to a local and transient effect under dynamic loading. The strain-rate effect and inertia effect were not analyzed. Jing et al. [19] have compared the impact mechanical properties of sandwich panels with different core layers, namely closed-cell aluminum foam, open-cell aluminum foam, and aluminum honeycomb, by a metallic foam projectile impact. A theoretical model was established to give a corresponding prediction in the dynamic response of sandwich panels. In that test, AFS appeared an obvious delamination at the lower gluing interface, which may result in severe consequences in applications. Liu et al. [17] have explored the dynamic mechanical behavior of AFSs with metal fiber laminate skins by an air gun impact test and finite element simulation. Experimental research was carried out to investigate the effect of the thickness of the core and face sheet on the impact response of the AFS and the failure modes. At a high-speed impact load, debonding between the core and face panel was a typical failure mode. And with the increase in thickness of the face sheet, the debonding ranges were wider. What was worth noticing is that the bottom face sheet of the sandwich structure often appears to interface debonding under the impact load, however, it is not clear whether the debonding is caused by the fracture and stretching of the bottom face panel, or the fracture of the lower face panel resulted in debonding. Thus, the dynamic mechanical behaviors of AFS are being researched indepth. Additionally, although cyclic loading is almost an authentic loading in most applications, the attention to the fatigue mechanical behavior of AFS is inconsistent with the degree of its importance. Few people have studied its fatigue properties. Harte et al. [20] have studied that AFS with aluminum face sheets were repeatedly loaded in a four-point bend to research the fatigue failure mechanisms. There were three failure modes: face fatigue, core shear, and indentation. An obvious debonding in large areas was not observed. Yan et al. [21] have explored the fatigue damage mechanisms of AFS with a carbon fiber face panel. When the fatigue loading was increased to 50%, an interfacial debonding came up and its initial location of debonding randomly appeared. To sum up, there is little research on the interface mechanics of AFSs, and the aforementioned analysis is almost based on its overall mechanical behavior. The interface plays a key role yet includes some flaws, namely easy delamination. Consequently, it is increasingly necessary to discuss the method of reinforcement of AFS interfacial mechanical properties.

There are two main kinds of interfacial connecting ways: metallurgy bonding and adhesive bonding, which would be considered to enhance the overall mechanical performance via reinforcing the interfaces. Welding, powder metallurgy, and other metallurgical methods can be used to fabricate AFSs with strong metallurgical bonding interfaces. In order to achieve metallurgical bonding, the welding method was one of the most direct approaches [22]. However, the filler metal is easy to oxide and so it needs an inert gas environment, which increases the cost and equipment requirements. Researchers from the Bremen Institute in Germany initially used the powder metallurgy method to prepare AFSs [23,24]. Powder metallurgy mainly uses molding or rolling to obtain a foamable precursor of the initial combination of the panel/core layer, and then the AFS is prepared through precise control of the foaming parameters. And this method can effectively connect the aluminum foam core with different metal face sheets and fabricate special-shaped components. Lin et al. [25] have investigated the fabrication of AFSs with mild steel face sheets via a vacuum-free diffusion bonding method, which is based on the powder metallurgy to prepare a foamable ALMg1/SiC_p_-based precursor. It realized the manufacture of a foam aluminum core and the bonding for the aluminum foam core and face plates. Although the bonding force exceeding the strength of the aluminum foam itself was obtained, this way is complicated and can only make specific panel materials and aluminum foam connect. A neotype friction stir welding (FSW) method was extensively carried out by Japanese researchers, and initially, this way was developed to bond aluminum alloy panels [26]. Hangai et al. [27,28] have used this method to fabricate aluminum foam/dense steel composites and then aluminum alloy foam core sandwich panels. Tensile tests on this AFS with porosity from 60 to 85% demonstrated that the fracture always occurred in the aluminum foam parts, not the welding interface. The downside of the metallurgy method is that it fails to fabricate large-size specimens. It can be seen from the research status for the metallurgical preparation of AFS that the metallurgical method has produced a firm bonding interface, however, this method is only applicable to the combination of aluminum foam and metal sheets rather than nonmetallic materials. It is tremendously urgent how to form higher quality aluminum foam sheet metal prefoaming body, and simplify the fabrication process so as to save on the costs of preparing AFSs. At present, adhesive bonding is one of the most widely used methods for producing AFSs due to its general applicability and simple process [29]. Many researchers have fabricated AFSs in this way and explored their related interface mechanical properties. Ruan et al. [30] have prepared AFSs bonded by SA80 epoxy adhesive and investigated the mechanical properties of AFS with aluminum alloy face panels under quasistatic indention loads, as well as discuss the effects of each parameter (face sheet thickness, core thickness, boundary conditions, and adhesive and surface condition of face sheets) on the energy absorption and mechanical response of AFSs. AFS with fiber-metal laminates as face sheets was also fabricated by a commercial epoxy resin [31]. Kwon et al. [32] have proven that the friction-surface-modifying and rolling process was a very effective method to enhance the bonding performance of aluminum foam by controlling its surface morphology. However, there are still some problems, such as adhesive joints cannot be disassembled without damage, adhesive cannot be used in high-temperature conditions, etc., which limit the use range of AFSs. Until now, the interfacial connection technologies of AFS are still continually explored, so that the requirements of high technology and high energy consumption can be met by further enhancing the interfacial connection strength of AFS to obtain a large-sized and multifunctional structure. With the development of materials science, nano, micro, and other microscale materials have brought new hope to the enhancement of polymer materials. It was carbon fiber that cannot be ignored among that. For the last few years, it was widely concerning to many correlative researchers that carbon fiber-reinforced polymer composites (CFRPs) had outstanding modulus and specific strength [33,34,35].

In the present study, short carbon fiber will be taken as a reinforcement phase to enhance the aluminum foam sandwich interface. Single-lap shear tests and three-point bending tests will be conducted to study the reinforcement performance. The effects of carbon fiber length and content will be considered. The failure of the mechanical and reinforcement mechanism of the reinforced interface will be studied in micro and macro. The detailed experiments and discussion are as follows. Figure 1 shows the fabrication process for short carbon fiber-reinforced aluminum foam sandwich structure (AFS) and the reinforcement mechanism of the reinforced interface.

## 2. Experimental Method

### 2.1. Materials

The 7050 matrix closed-cell aluminum foam block was selected as the sandwich structure core layer. The 6061 aluminum alloy panel was chosen as a face sheet material since aluminum alloy was universally thought of as an appropriate face sheet material [15,20]. The epoxy resin type was E44-bispheno1-A (Hu’nan Baxiongdi New Material Co., Ltd., Changsha, China) and the curing agent was 650 resin firming agent (Hu’nan Baxiongdi New Material Co., Ltd. Changsha, China). The carbon fiber bundle (CFS-II-300, CARBON Technology Group Co., Ltd., Chengdu, China) was cut into lengths of 3 mm, 6 mm, and 9 mm. The relevant mechanical parameters for the above materials were shown in Table 1.

In single-lap shear tests, the low tensile strength of the aluminum foam itself was prone to fracture, making it difficult to obtain accurate data on the bonding interface’s mechanical properties. Hence, the 6061 aluminum alloy blocks were selected by drilling holes in the surface to simulate aluminum foam blocks. In order to ensure that the surface of the aluminum block and aluminum foam have similar surface characteristics and internal morphology, we punched the aluminum block according to the diameter and depth of the aluminum foam surface hole. In the subsequent experiments and tests, the surface morphology of the punched aluminum block and foam aluminum was all polished with 600 mesh sandpaper to ensure the similarity of the interface, and the surface of the aluminum block corrosion using hydrofluoric acid to simulate the internal roughness of aluminum foam, as shown in Figure 2.

### 2.2. Fabrication of Single Lap Joints (SLJs) and Aluminum Foam Sandwich Structures (AFSs)

Drilled 6061 aluminum alloy blocks were etched by hydrofluoric acids for 20 min to obtain smooth faces. They were immersed in an appropriate sodium bicarbonate solution and then washed out with deionized water. In this study, sandpaper with 600 particles per square inch was applied to polish the bonding faces, including the faces of drilled aluminum alloy blocks, aluminum foams, and 6061 aluminum alloy sheets. What was testified as a valid method to increase surface roughness by many previous researchers [36,37] is using sandpaper to polish surfaces.

The chopped carbon fiber bundle was added into acetone with an ultrasonic environment for exceedingly dispersing the fiber bundle into fibril. For the sake of removing pollutants and sizing agents thoroughly, the CFs were continually washed with acetone in a Soxhlet extractor. The CFs were put into the appropriate concentrated HNO_3_ to introduce carboxyl groups (−COOH) and hydroxyl groups (−OH), respectively.

Short carbon fiber reinforced adhesives were prepared first. The acetones with a total weight of 15% were added into epoxy resin to easily and fully disperse the modified carbon fibers. The short carbon fibers were mixed into the above hybrid and stirred till it was completely separated and dispersed. The visual checking of the short fiber dispersion was sufficient and reliable. Furthermore, in order to remove the bubbles produced in the stirring process, the mixture was put in a vacuum-degassing oven for 30 min at the vacuum pressure of 0.8 bar at 30 °C. Finally, resin-firming agents with a curing agent of equal weight ratio (1:1) were dispersedly mixed into the above hybrid. The adhesive with short carbon fiber was ready.

After that, single lap joints (SLJs) were fabricated utilizing the reinforced adhesives respectively consisting of modified short carbon fibers in four total weight percentages of 0.1, 0.2, 0.3, and 0.4, and three lengths of 3 mm, 6 mm, and 9 mm. So, the certain length and content of short carbon fibers added into adhesives were grouped as shown in Table 2. In order to obtain a uniform adhesive layer thickness, a tailored fixture illustrated in Figure 3 was able to support an invariable pressure. The SLJs were cured for 2 h at 80 °C in an air blower (the optimum curing temperature [21]), then for exceeding 48 h at room temperature, which could insure sufficient curing.

Furthermore, the additive amount of short carbon fiber in AFS was based on the optimal comprehensive performance of SLJ in single lap shear tests. Finally, the AFSs were manufactured whose fabrication condition and process were identical to the SLJ’s.

### 2.3. Single Lap Shear Test and Three-Point Bending Test

The dimensions of the single lap joint (SLJ) employed for experimental research are illustrated in Figure 4. The SLJ samples were tested using an electronic universal tensile testing machine (WDW3100) with a stable movement rate of 1 mm/min until it was completely destroyed. Fracture surfaces were observed through a scanning electron microscope.

Specimens of the sandwich panel were loaded in a three-point bending configuration, as shown in Figure 5 WDW3100 electronic universal tensile testing machine. The indenter moved at a rate of 2 mm/min to indent the specimen at the midpoint of the top face sheet. The diameter of the cylindrical indenter was 10 mm. In the same way, the specimen was supported on two support pins with a diameter of 10 mm. The dimension of the sandwich panel was shown in Figure 5, and its width was 30 mm. The deformation of foam cores and both face sheets during tests was recorded by a digital camera.

## 3. Result and Discussion

### 3.1. Single-Lap Shear Behavior of the Reinforced Interface

#### 3.1.1. The Load-Displacement

Generally, the load-displacement curve of a single lap joint was artificially divided into four stages: (1) the tightening stage, the curve grows briefly and then experiences a short platform; (2) the nonlinear rise stage, the curve grows in volatility; (3) the elastic deformation stage, the curve grows linearly; and (4) the failure stage, the load experienced a cliff descent. As shown in Figure 6, the load-displacement curve of the reinforced interface was contrasted with that of the unreinforced interface under single lap shear tests. The load-displacement curve of the reinforced interface was chosen for further analysis. From points O to A, this curve experienced a tightening stage, which made the specimen support force. After point A, the joint started to withstand shear force. From point A to B, as the displacement increased, the load presented a nonlinear rise. This might be caused by the initial elasticity of the bonding interface. The curve almost linearly increased from point B to C, and it was a process of uniform stress on the joint. When the curve reached the load top point (point C), a devastating break occurred, which meant the joint failed. The peak load of the reinforced interface was 6.28 kN and that of the unreinforced interface was 3.54 kN. The peak load of the reinforced interface increased by 77.40%, which powerfully illustrated that the addition of carbon fiber enhanced the bonding strength.

#### 3.1.2. Effects of Carbon-Fiber Length and Content on the Mechanical Behavior of the Gluing Joint

The reinforcement effect of carbon fiber on the mechanical properties of the gluing interface was related to the content and length of the fiber so the effects of carbon fiber content and length were studied. As shown in Table 3, it was the shear strength of each specimen in the group. The bonding interface with CF was enhanced to different degrees, except for the case that the overall effect was reduced due to the bonding process. The group with the highest shear strength was group M (0.4 wt%, 9 mm CF), whose shear strength reached 16.84 MPa. The maximum value of a single specimen also appeared in group M, whose maximum value was 19.25 MPa. However, the test value of this group was relatively discrete and had poor stability. The possible reason was that in the stirring process, the dispersion was not uniform. The enhancement effect of group E (0.2 wt%, 3 mm CF) was second only to that of group M, with the enhancement degree reaching 73.65% and the average shear strength reaching 16.41 MPa. And the data in the group were relatively stable, considering that the shorter carbon fiber had a more uniform and stable dispersion in the stirring process.

As shown in Figure 7a, the effect of the CF addition amount on the shear strength of the enhanced bonding interface was studied. When the CF length was 3 mm and the content was 0.2 wt%, the shear strength reached the peak value of 16.41 MPa. When the CF length was 6 mm, the enhancement effect was stable, and the shear strength was increased by an average of 5.22 MPa. When the length of the CF was 9 mm, the statistical rule of the effect of the additive amount was enhanced first and then weakened when the additive amount was less than 0.4 wt%, and a peak occurred when the additive amount was 0.2 wt%, and the overall strength was enhanced by 54.92% compared with that of pure adhesive. However, when the content was 0.4 wt%, the overall strength had been greatly enhanced, unexpectedly. It was considered that the length of CF reached a certain level, which affected the overall strengthening effect.

As shown in Figure 7b, the effect of CF length on the shear strength of the enhanced interface was investigated. The shear strength reached the peak, 15.15 MPa when CF content was 0.1 wt% and CF length was 6 mm. When the CF content was 0.2 wt%, the shear strength reached the peak at 16.41 MPa when the CF length was 3 mm, and then showed a weak strengthening effect when the CF length was 6 mm and 9 mm. When the content of CF was 0.3 wt% and the length was 6 mm, the peak shear strength was 15.28 MPa. When the content of CF was 0.4 wt%, the effect of CF addition on the enhancement effect gradually increased, and the maximum value was reached when the CF length was 9 mm.

Through the above analysis, the ratio of two parameters (carbon fiber content and length) with a relatively stable and high strengthening effect was obtained. Group E (0.2 wt%, 3 mm of short CF) was selected to carry out the next quasistatic three-point bending experiment to verify and analyze the strengthening effect of the carbon fiber reinforced aluminum foam sandwich panel.

### 3.2. Three-Point Bending Behavior of the AFS with Short Fiber Reinforced Interface

Three-point bending tests on the aluminum foam sandwich structure (AFS) were conducted to study the reinforced effect of the reinforced interface on the overall mechanical properties. The enhancement effect was understood by comparing the AFS with a short fiber-reinforced interface and the AFS without reinforcement under a three-point bending load.

#### 3.2.1. The Load-Displacement Curve and Deformation Process

It is by now universally accepted that the load-displacement curve of AFS was artificially divided into three stages [10]: (1) the elastic deformation stage, the curve grows linearly; (2) the yield stage, load curve shows a nonlinear variation, and holes destroying and collapsing occurred to absorb energy, which is the most time lasting phase than others; and (3) the failure stage, the load experienced a cliff descent.

Figure 8 showed the load-displacement curve of AFS prepared by reinforced interface and unreinforced interface. In the b-c/b_0_-c_0_ stage, the force-displacement curve of the unreinforced AFS was very short, which was due to the lack of bonding interface strength, which lead to the degumming of the upper panel and the core layer. However, further pressing of the indenter described in the b-c stage of the short carbon fiber reinforced AFS did not result in the degumming of the upper panel and core layer, so the displacement was longer, and the load was larger, in this stage. The unreinforced AFS was in the degumming stage at the c_0_-d_0_ stage, and the effective displacement was short, while the short carbon fiber reinforced AFS was in the process of crack propagation and core energy absorption at the c-d stage, and the effective displacement was much greater than the former. In the d-e stage of the reinforced AFS, the interface went through a long platform area during the compression deformation, and the overall failure occurred after it reached the e point due to the large deformation. Without the reinforced AFS d_0_-e_0_ stage, due to the presence of degumming in the last stage, this one-phase load fell faster, and the bonding interface strength was not enough, with the panel and core layer degumming again, causing the overall structural failure quickly, so the overall effective displacement than short carbon fiber reinforced aluminum foam sandwich panel was much lower.

The deformation process of the carbon fiber reinforced AFS in Figure 9 corresponded to the inflection point of the red force-displacement curve in Figure 8. Point a was the initial position of the three-point bending. From point a to b, the structure was under overall force, and this stage was the linear elastic deformation stage. Point b was the first yield point where the local area of the structure reached the yield stress. After point b, there was a weak increase in the load until it reached the peak at point c. A crack was generated in the core layer at stages b and c, and the crack source was located near the large hole under the foam with the maximum bending radius. When the bending process was from points c and d, obvious upper indenter compression and plastic hinge deformation could also be observed in Figure 9d, where the plastic hinge was caused by the bending moment on the section reaching the plastic limit bending moment, which led to rotation. When the bending moment on this section was less than the ultimate plastic bending moment, rotation was not allowed. Therefore, the plastic hinge could transmit a certain bending moment. Stress was concentrated in the area under the indenter. It could be observed that crack propagation caused by core layer shear e in Figure 9d on the lower panel led to a continuous decrease in the load before degumming occurred at the position of yellow arrow ① in Figure 9d. The cavity plastic deformation in stages d and e was approximately the plateau area where the AFS absorbed energy in the form of cavity collapse, which was a typical phenomenon caused by the existence of aluminum foam. As compression progresses, core layer shear ② in Figure 9e continued. Due to inconsistent deformation, degumming occurred at the yellow arrow ② in the figure, which also led to the direction of degumming opposite to that of the unreinforced aluminum foam sandwich beam. From points e to f, the core layer was inconsistent with the panel, which lead to the degumming of the lower panel and core layer at the position shown by the yellow arrow ③ of Figure 9f, resulting in rapid failure of the structure.

To be concrete, as shown in Table 4, the peak load of AFS without a reinforced interface was dropped by 0.48 kN and its effective displacement was less by 4.92 mm. As a result, the energy absorption capacity was greatly enhanced. The energy absorption of AFS with a short fiber-reinforced interface improved by 125.95%, which was an encouraging result.

The three-point bending mechanical behavior of the present reinforced interface can be compared with that of the metallurgical interface. Wang et al. [38] have proposed a new fiber metal interlayer composite structure to manufacture AFS in Figure 10. Compared to normal AFS, the new composite structure improved the energy absorption capacity by increasing the effective displacement, but its maximum carrying capacity decreased. The present study results (as shown in Figure 8) could exceed the new composite structure since the reinforced method in the present study possessed a higher peak load and bigger energy absorption region than both. Additionally, Zu et al. [39] improved the effective displacement and maximum bearing capacity by means of the metallurgical method, as shown in Figure 11a,b. Combined with the results in Figure 8, a heart-stirring conclusion can be drawn that the mechanical properties of the short fiber-reinforced interface in the present study could be compared to those of the metallurgical interface.

#### 3.2.2. The Failure Mode

Evidently, different load-displacement curves brought different failure modes. There were three failure modes in terms of adhesive bonded AFS in previous research: yield damage of both core and bottom face-sheet (Failure mode I), yield damage of foam core (Failure mode II), and debonding between the adhesive interface (Failure mode III) [40].

Figure 12a showed the failure diagram of the unreinforced bonding AFS. Based on the analysis of the above deformation process, the failure was mainly caused by the weak adhesive force between the panel and the foam core. At the beginning of plastic deformation, the longitudinal shear force caused by the indenter pressing was greater than the bonding strength between the panel and the core, leading to degumming. The following core layer fracture and final failure were also due to the rapid failure of the overall structure caused by an insufficient bonding strength between the lower panel and core layer. Therefore, the failure mode of the unreinforced AFS was mainly an adhesive interface failure (failure mode III). Figure 12b showed the failure mode diagram of the carbon fiber-reinforced AFS. Combined with the deformation process, there were three main failure modes of the structure: (1) upper indenter indention, (2) aluminum foam core layer shear, and (3) plastic hinge deformation.

### 3.3. Enhancement Mechanism of the Interface

The results of the single lap shear test and three-point bending test demonstrated that short carbon fiber could effectively improve the integral properties of aluminum foam sandwich structure (AFS). The reinforcement mechanism was investigated via a scanning electron microscope and a macro analysis, as shown in Figure 13 and Figure 14.

Three kinds of enhancement mechanisms can be found in the present study. First is the pull-off and pull-out of short carbon fibers. As shown in Figure 13, due to the short fibers being added, the fiber fracture or pull out at the bonding interface under the load, which effectively improved the tangential strength. At the same time, the short carbon fiber prevented the further expansion of cracks in the epoxy resin, and also improved the bonding interface strength. Second is the adhesion effect. As shown in Figure 14a1–d1, the addition of short carbon fiber increased the effective bonding area on both the face of the aluminum foam and the 6061 aluminum alloy face sheet. Before adding the short carbon fiber, the pore edge had mostly no adhesive adhesion. After adding the short carbon fiber, due to the aggregation of carbon fibers, the adhesive adhesion of the hole edge was better, which made the failure mode from adhesion to cohesion. That resulted in the improvement of adhesive strength. Third, the mechanical self-locking effect. In Figure 14a–d, blue dotted circles can illustrate that the carbon fiber enhanced the adhesive filling performance of aluminum foam holes. Because the carbon fibers broke the surface tension of epoxy resin. Hence, it improved the mechanical self-locking effect of the bonding interface.

## 4. Conclusions

This study adopted carbon fiber which was modified by liquid cyclohexane oxidation to reinforce the interface of the aluminum foam sandwich structure (AFS). The single lap shear tests were conducted to research the effects of different lengths and content of carbon fiber added into adhesive. Three-point bending tests were implemented to scientifically terrify the enhancement performance. Then the reinforcement mechanism was analyzed via the optical microscope and the scanning electron microscope. Several conclusions can be summarized as follows:The result of the single lap shear test indicated that the modified carbon fiber had a positive effect on the shear strength of the interface of AFS. The single lap shear reinforced with 3 mm length and 0.2 wt% content carbon fiber showed the highest enhancement, increasing by 73.65%.Three-point bending tests proved that the strength of the short fiber reinforced AFS was higher than the unreinforced AFS. There was a giant improvement in energy absorption. The numerical value was from 17.49 (kN × mm) to 39.51 (kN × mm) whose growth rate reached 125.95%. The problem of failure of AFS due to degumming was improved.The reinforcement mechanism of the carbon fiber reinforced interface for AFS was due to the influence of the carbon fiber on adhesive interface shear failure. Three categories of reinforcement mechanisms were discovered in this study: (a) the pull off and pull mechanism: the modified carbon fiber performed as the bridge, the bonding strength improved because of the pull off, and pull out of fibers; (b) the adhesion effect: the carbon fiber gathered in the hole edge resulted in epoxy resins being gathered in there too, which increased the effective bonding area of the interface; (c) the mechanical self-locking effect: the carbon fiber enhanced the adhesive filling performance of aluminum foam holes, which improved the mechanical self-locking effect of the bonding interface.

In conclusion, the present work provided an effective method to gain AFS with a high interfacial mechanical performance by gluing method, the reinforced effect can be compared to that of the interface prepared by the metallurgy method.

## Figures and Tables

**Figure 1 materials-15-09012-f001:**
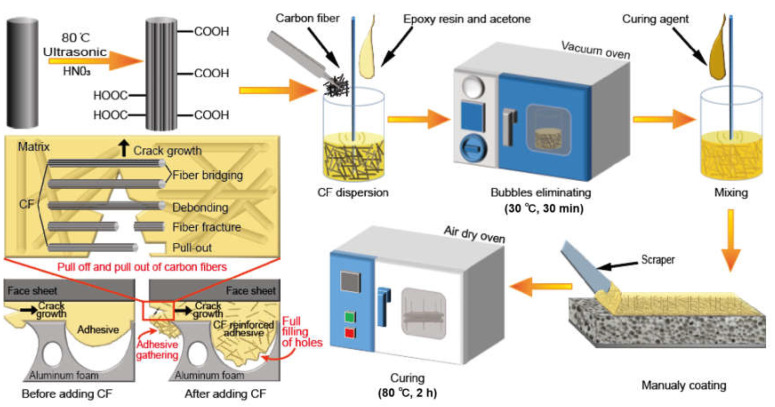
Schematic illustration of the fabrication process for short carbon fiber reinforced Aluminum Foam Sandwich Structure (AFS) and reinforcement mechanism of the reinforced interface.

**Figure 2 materials-15-09012-f002:**
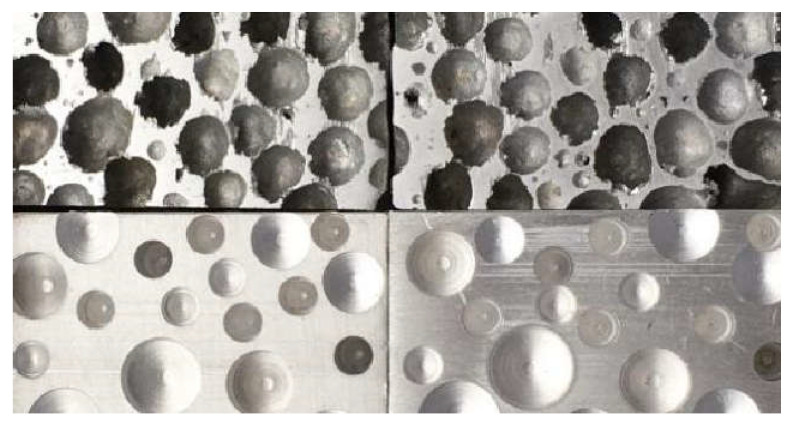
Holes comparison of the aluminum foam and aluminum-foam-like.

**Figure 3 materials-15-09012-f003:**
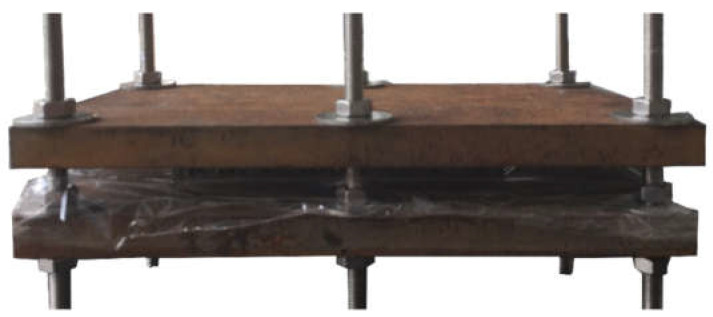
The fixture utilized for the fabrication of SLJs and AFSs.

**Figure 4 materials-15-09012-f004:**
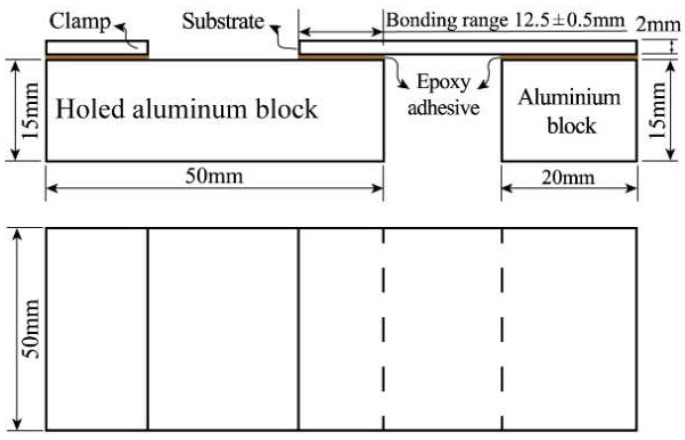
The geometry and dimensions of an SLJ.

**Figure 5 materials-15-09012-f005:**
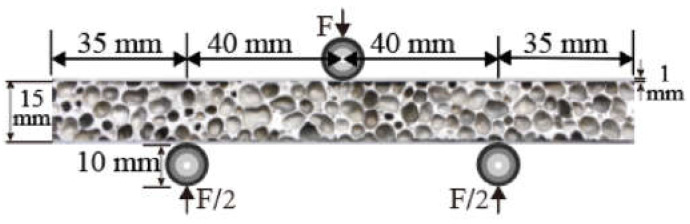
Schematic diagram of three-point bending test.

**Figure 6 materials-15-09012-f006:**
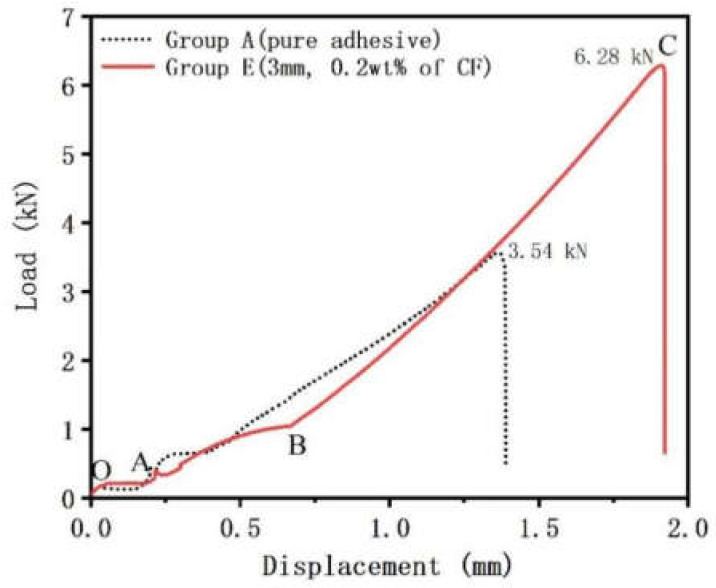
Typical load-displacement curves of group A (pure adhesive) and group E (3 mm, 0.2 wt% of CF).

**Figure 7 materials-15-09012-f007:**
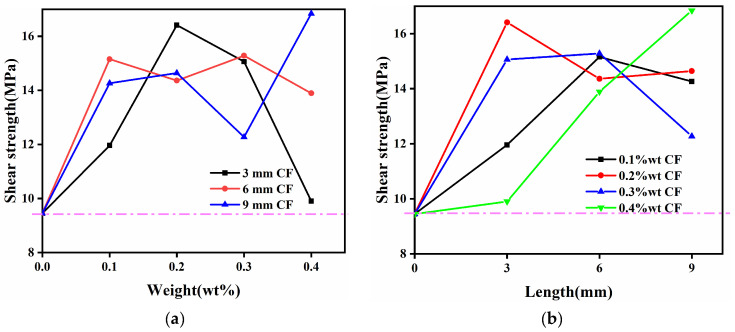
Shear strength changed with the added CF weight percentage (**a**) and the length (**b**).

**Figure 8 materials-15-09012-f008:**
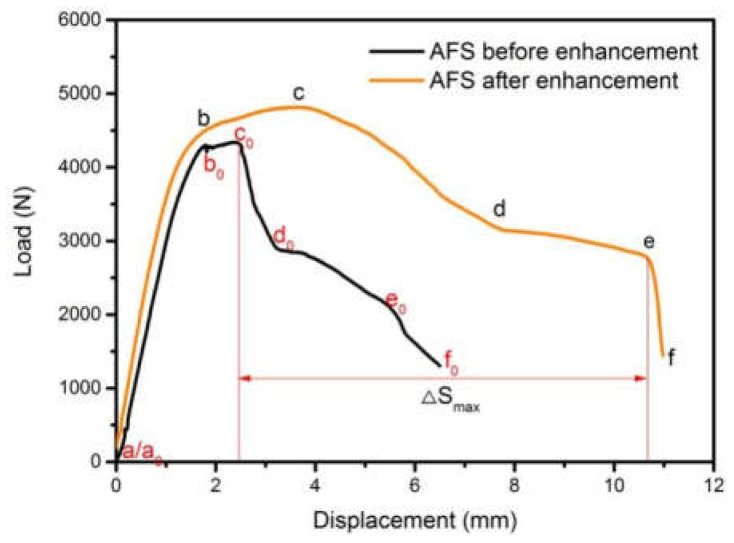
Load-displacement curves of AFS before and after interface enhancement.

**Figure 9 materials-15-09012-f009:**
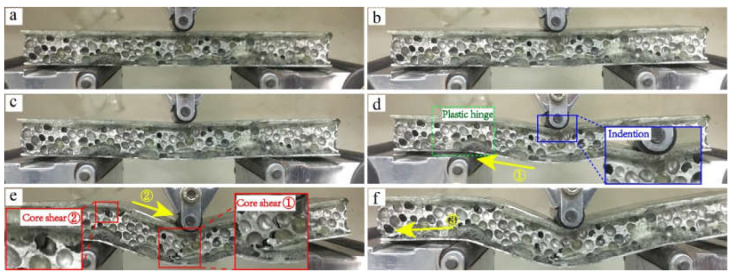
Deformation process from (**a**) to (**f**) of AFS after interface enhancement under three-point bending.

**Figure 10 materials-15-09012-f010:**
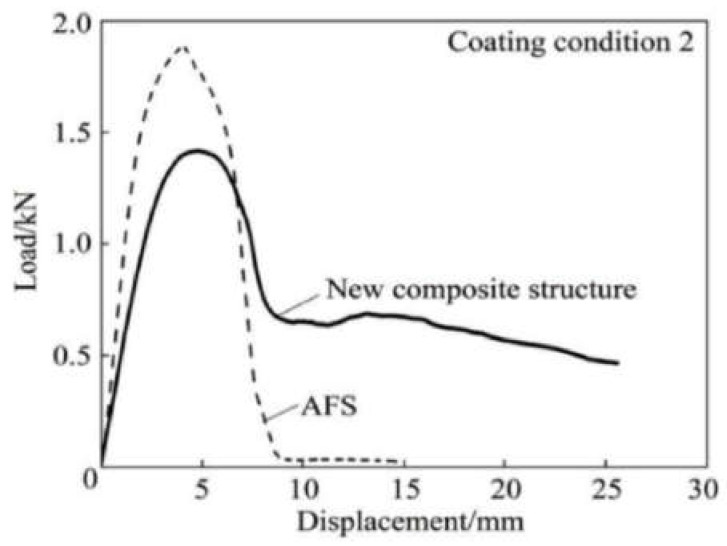
Load-displacement curves of new composite structure and AFS coated with epoxy resin [38] (Figure 10 is copied from other published papers, if original pictures were needed, please check the references.).

**Figure 11 materials-15-09012-f011:**
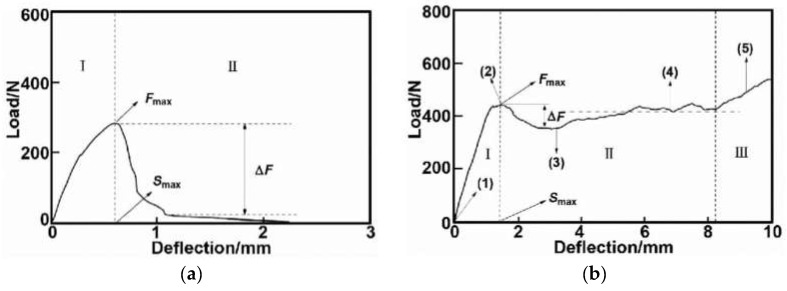
Load-displacement curves of (**a**) adhesive bonded AFS [39] and (**b**) metallurgical bonding AFS [39] (Pictures a and b are copied from other published papers, if original pictures were needed, please check the references.).

**Figure 12 materials-15-09012-f012:**
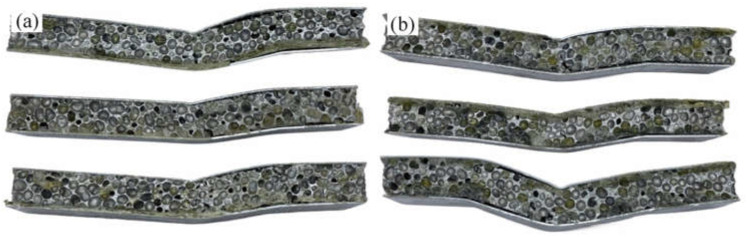
Macroscopic graph of (**a**) failed unreinforced AFS and (**b**) failed short CF reinforced AFS.

**Figure 13 materials-15-09012-f013:**
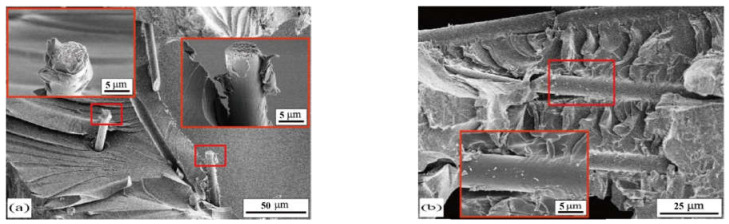
SEM images of fracture surfaces of (**a**) carbon fiber breaks in the vertical direction (**b**) adhesive adhering to the side of the carbon fiber.

**Figure 14 materials-15-09012-f014:**
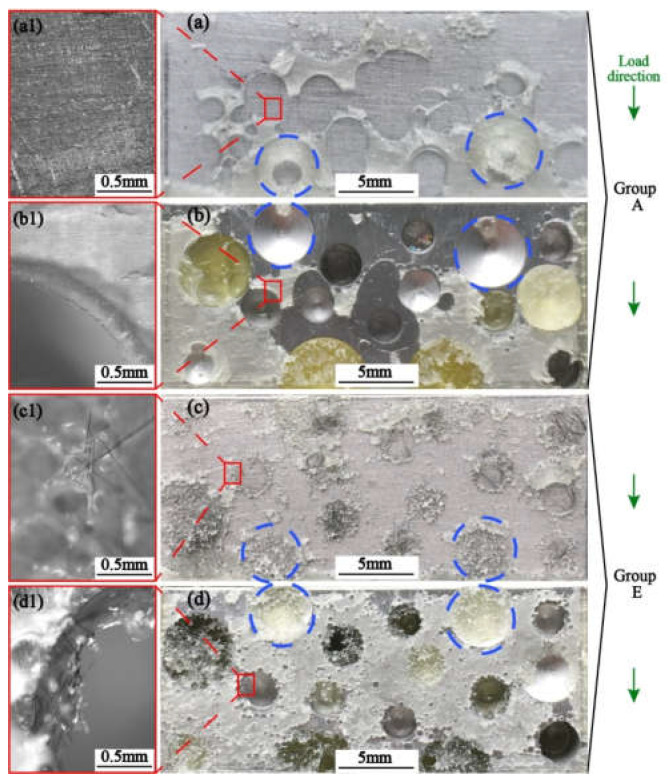
Fracture surfaces of unreinforced SLJ (Group A, pure adhesive) with (**a**) panel, (**b**) the perforated aluminum block and reinforced SLJ (Group E, 0.2%wt & 3 mm CF) with (**c**)panel, (**d**)the perforated aluminum block.

**Table 1 materials-15-09012-t001:** Mechanical parameters of different materials in this study.

Material	Density	Average Pore Size /mm	Pore Size Range /mm	Tensile Strength /MPa	Yield Strength /MPa	Elongation /%	Tensile Modulus/MPa
Aluminum foam	0.60 g/cm^3^	3.77	1.66~6.67	-	4.4	-	-
6061 aluminum alloy	2.75 g/cm^3^	-	-	333	300	-	-
Carbon fiber bundle	297.1 g/m^2^	-	-	3298.9	-	1.5	2.5 × 10^5^

**Table 2 materials-15-09012-t002:** Length and content of carbon fiber of different groups.

	Length	0 mm	3 mm	6 mm	9 mm
Content					
0.0%wt		A	--	--	--
0.1%wt		--	B	C	D
0.2%wt		--	E	F	G
0.3%wt		--	H	I	J
0.4%wt		--	K	L	M

**Table 3 materials-15-09012-t003:** Shear strength of each specimen.

	Number	1 /MPa	2 /MPa	3 /MPa	Average /MPa	Max /MPa	Standard Deviation/MPa	Enhance Degree/%
Group								
A (pure adhesive)		9.48	9.43	9.45	9.45	9.48	0.025	-
B (0.1 wt% & 3 mm CF)		10.18	12.59	13.10	11.96	13.10	1.56	26.56
C (0.1 wt% & 6 mm CF)		15.46	15.46	14.52	15.15	15.46	0.54	60.32
D (0.1 wt% & 9 mm CF)		15.04	13.21	14.54	14.26	15.04	0.95	50.90
E (0.2 wt% & 3 mm CF)		16.75	16.02	16.46	16.41	16.75	0.37	73.65
F (0.2 wt% & 6 mm CF)		13.34	15.42	14.33	14.36	15.42	1.04	51.96
G (0.2 wt% & 9 mm CF)		14.08	14.13	15.72	14.64	15.72	0.93	54.92
H (0.3 wt% & 3 mm CF)		14.15	16.39	14.63	15.06	16.39	1.18	59.37
I (0.3 wt% & 6 mm CF)		14.74	16.54	14.55	15.28	16.54	1.10	61.69
J (0.3 wt% & 9 mm CF)		12.63	12.57	11.61	12.27	12.63	0.57	29.84
K (0.4 wt% & 3 mm CF)		10.69	9.45	9.55	9.90	10.69	0.69	4.76
L (0.4 wt% & 6 mm CF)		15.15	13.83	12.70	13.89	15.15	1.22	46.98
M (0.4 wt% & 9 mm CF)		17.11	19.25	14.15	16.84	19.25	2.56	78.20

**Table 4 materials-15-09012-t004:** Peak load, displacement, effective displacement and energy absorption of each specimen.

Specimen Name	Peak Load /kN	Displacement of Peak Load/mm	Effective Displacement/mm	Energy Absorption /kN × mm
Unreinforced AFS	4.34	2.40	5.77	17.49
Reinforced AFS	4.82	3.59	10.68	39.51

## Data Availability

All data have been elaborated in the text with no other data.

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
