# Peer review of "The Mechanical Behavior and Enhancement Mechanism of Short Carbon Fiber Reinforced AFS Interface"

_materials, 2022, doi:10.3390/ma15249012_

Round 1

Reviewer 1 Report

1.      The reproduction of the surface of the foam in the block of aluminum alloy raises doubts, however, because the surface structures differ significantly.

2.      How was the value of the pressure force (pressure) during gluing determined in order to obtain the same thickness of joints in all samples?

3.      The results presented in Table 3 lack, for example, the standard deviation of the mean value, which would present the dispersion of the results.

4.      Please try to explain the mechanism of reinforcement in tensile strength tests, because the graphs and data do not show a clear relationship between the length of the fibers and its weight share in the adhesive on the strength of the joints??

5.      Conclusions 1 and 2 are too detailed, they should be generalized or relative values should be used.

Author Response

Thank you very much for your review. The specific reply and modification are in the attachment, please check it. If there is any questions, plaease contact us. Thank you.

Reviewer 2 Report

In this paper, the authors have introduced the idea of adding the short carbon fiber to the epoxy resin to overcome the shortcoming of bonded interface in aluminum foam sandwiches. Single lap shear tests and three-point bending tests have been conducted out to study the effects of different length and content carbon fiber added into adhesive. The subject of this study is interesting. However, the reviewer has some comments that should be addressed before publication.

Comments:

1) The grammatical and writing errors are available in the manuscript. For example, page 2 line 8, some words are bolded.

2) In Fig. 7b, shear strength of the sample with 0.4%wt CF increases with increasing the length of carbon fibers. However, the samples with other values of CF weight percentages display different behavior. What is the reason?

3) Fig. 10a is repetitive. It is shown previously in Fig. 8.

4) Different parts of Fig. 10 are not relevant together. It is recommended that these parts to be displayed in separated figures.

Author Response

(The authors gave the same response as above.)

Round 2

Reviewer 2 Report

The authors have considered all reviewer's comments in the preparation of the revised version of the paper. It is recommended for publication at the present format.